# Characteristics of 21 Patients with Secondary Hemophagocytic Lymphohistiocytosis—Insights from a Single-Center Retrospective Study

**DOI:** 10.3390/medicina61060977

**Published:** 2025-05-26

**Authors:** Radosław Dziedzic, Stanisława Bazan-Socha, Mariusz Korkosz, Joanna Kosałka-Węgiel

**Affiliations:** 1Jagiellonian University Medical College, Doctoral School of Medical and Health Sciences, św. Łazarza 16, 31-530 Kraków, Poland; radoslaw.dziedzic@doctoral.uj.edu.pl; 2Jagiellonian University Medical College, Department of Internal Medicine, Faculty of Medicine, Jakubowskiego 2, 30-668 Kraków, Poland; stanislawa.bazan-socha@uj.edu.pl; 3Jagiellonian University Medical College, Department of Rheumatology and Immunology, Jakubowskiego 2, 30-688 Kraków, Poland; mariusz.korkosz@uj.edu.pl

**Keywords:** hemophagocytic lymphohistiocytosis, haemophagocytic syndrome, macrophage activation syndrome, hemophagocytosis

## Abstract

*Background and Objectives*: Hemophagocytic lymphohistiocytosis (HLH) is a rare hyperinflammatory condition characterized by excessive activation of cytotoxic lymphocytes and macrophages, resulting in a cytokine storm, multiorgan damage, and high mortality. HLH is classified into primary (genetic) and secondary (acquired) forms, with diagnosis often challenging due to nonspecific symptoms. Macrophage activation syndrome (MAS) refers to the secondary HLH triggered by rheumatic diseases. In this study, we retrospectively analyzed the clinical and laboratory features of patients with secondary HLH to enhance understanding of this life-threatening condition and summarize emerging management strategies. *Materials and Methods:* This single-center retrospective study analyzed medical records of patients hospitalized with HLH at the University Hospital in Kraków, Poland, from 2013 to 2024, based on HLH-2009 criteria and HScore > 169 points. Diagnostic criteria included clinical, laboratory, and histological findings, e.g., hemophagocytosis in bone marrow, circulating cytopenia, and elevated ferritin levels. *Results*: A total of 21 patients met the criteria for HLH diagnosis, with a median age of 35 (range: 19–67) years, including 12 women (57.1%). The median HScore among the patients was 244 (range: 208–304) points. Fever was the most common presenting symptom, occurring in all cases. High ferritin, hypertriglyceridemia, and hypofibrinogenemia in peripheral blood were also prevalent. Bone marrow hemophagocytosis was confirmed in 66.7% of cases (*n* = 12/18 of available data). Regarding immunosuppressive therapy, glucocorticosteroids were the most frequently used (used in all cases). Four (19.0%) patients died during HLH (cases triggered by lymphoma [twice], Epstein–Barr virus infection, unknown reason). Compared to survivors, these patients had lower counts of white blood cells, neutrophils, and lymphocytes at diagnosis (*p* < 0.05 for all). *Conclusions*: Secondary HLH is a severe syndrome requiring rapid diagnosis and timely intervention to improve patient outcomes. Lower white blood cell, neutrophil, and lymphocyte counts present worse prognostic factors.

## 1. Introduction

Hemophagocytic lymphohistiocytosis (HLH), also known as hemophagocytic syndrome, is a rare hyperinflammatory condition typified by persistently activated cytotoxic lymphocytes and macrophages, leading to a “cytokine storm” [1]. HLH manifests in a spectrum of clinical presentations ranging from persistent fever, hepatosplenomegaly, cytopenias, elevated ferritin level, and disseminated intravascular coagulopathy to multiorgan dysfunction, posing significant diagnostic and therapeutic challenges, particularly since untreated can rapidly lead to death [2,3]. It is divided into primary (an inborn error of immunity) and secondary (acquired mechanisms) forms. Primary HLH arises from pathological gene variants affecting immune regulation pathways, such as those involved in perforin or cytotoxic granule release. At the same time, secondary HLH may develop in response to infections, malignancies, autoimmune diseases, immunodeficiency, and immunosuppressive or cytotoxic therapies; in some cases, the etiology is unknown [4]. Interestingly, macrophage activation syndrome (MAS) is a specific term for HLH occurring due to rheumatic diseases, usually systemic juvenile idiopathic arthritis or autoinflammatory syndromes [5]. However, independently of causes, HLH is characterized by excessive activation of immune and inflammatory cells, leading to persistent multisystem inflammation [6]. Key players in its pathogenesis include macrophages, histiocytes, and CD8+ T cells, which fail to eliminate activated macrophages. Consequently, activated cells release large amounts of numerous cytokines responsible for multiorgan damage, such as IFN-γ, CXCL9, and CXCL10, whose levels correlate to some parameters of HLH severity [7]. Next, other immunologic parameters specific to HLH include high levels of the soluble IL-2 receptor (sIL-2R, CD25) and low count or even absence of NK cells in circulating blood [8].

Nevertheless, the final diagnosis of HLH might be challenging, especially in case of a complex clinical picture, including signs of initial illness [9]. Moreover, the relatively high mortality of HLH patients due to several reasons, including bleeding to the visceral organs, opportunistic infection, or multiple organ failure, points out the need for further investigations to improve patient-centered outcomes [10]. Thus, timely diagnosis and optimal treatment strategy are paramount for the final prognosis. However, data regarding the clinical course, management, and outcomes of adult secondary HLH forms are still scarce. Nowadays, research is focused on unifying nomenclature and providing validated criteria for diagnosis of HLH across specialties, as well as characterizing epidemiology, contributors, management practices, and outcomes in these patients [11].

Therefore, we decided to perform a retrospective analysis of clinical and laboratory characteristics and treatment patterns in secondary HLH patients treated in our hospital during the last decade. Due to the rarity of the syndrome, we believe every report is valuable in understanding this life-threatening condition.

## 2. Patients and Methods

### 2.1. Study Population

This is a single-center retrospective study in which we reviewed the medical records of HLH patients hospitalized from 2013 to 2024 at the University Hospital in Kraków, Poland. We used the HLH-2009 diagnostic criteria [12]. We summarized the criteria stated by the Histiocyte Society below (Table 1).

In all cases, we collected clinical and laboratory data, such as demographic (age and sex), clinical (accompanying illness followed by HLH; with time of its beginning to HLH symptoms), and treatment patterns (glucocorticosteroids and other immunosuppressive drugs). Furthermore, we analyzed laboratory parameters such as complete blood cell count, liver enzymes, lactate dehydrogenase, albumin, triglycerides, fibrinogen, serum ferritin, D-dimer, NK cells, and C-reactive protein at diagnosis.

The cytopenia was stated in case of affecting at least two lines: hemoglobin < 9 g/dL, platelets < 100 × 10^3^/μL, and/or neutrophils < 1000/μL. The NK dysfunction means an absence or very low number of NK cells. NK cells were identified in peripheral blood as CD3^−^CD56^+^ lymphocytes by flow cytometry, and the results are presented as percentages and absolute counts. Hemophagocytosis was typically studied in bone marrow; however, in selected cases, also in lymph nodes, spleen, or cerebrospinal fluid, depending on the main clinical presentation. Elevated ferritin was diagnosed when the serum level exceeded 500 ng/mL. Hypertriglyceridemia, when the level of triglycerides was above 265 mg/dL. Hypofibrinogenemia, when the level of fibrinogen was below 1.8 g/L. Hepatitis was defined as elevated levels of either aspartate transaminase or alanine transaminase (above 1.5 x upper limit of the norm). Hyponatremia was defined as a natrium blood level below 135 mmol/L. Regarding the clinical picture, splenomegaly was diagnosed based on physical examination and confirmed by ultrasound or computed tomography when spleen size exceeded age-specific norms. Lymphadenopathy was defined as lymph nodes exceeding 1 cm in the largest dimension, detected clinically or by imaging. Neuropsychiatric symptoms included altered consciousness, seizures, or cognitive disturbances, assessed clinically and with magnetic resonance or using cerebrospinal fluid analysis when indicated.

We also calculated the HScore in each patient. This score was employed to diagnose reactive hemophagocytic syndrome. It includes clinical (fever and organomegaly) and laboratory (cytopenia, elevated ferritin, hypertriglyceridemia, hypofibrinogenemia, liver damage, and hemophagocytosis in bone marrow aspirate) features of HLH and a history of immunosuppression use in the past [13]. We have provided a summary of parameters included in the Hscore (Table 2).

The final diagnosis of HLH may be stated in patients meeting the HLH-2009 criteria and with the HScore above 169 points. Regarding this score, in cases with a total score of >169 points, the probability of HLH is >99% [13].

In patients with underlying rheumatic diseases, cases fulfilling HLH diagnostic criteria were further classified as MAS (macrophage activation syndrome), in accordance with the current understanding that MAS represents a subtype of secondary HLH occurring in the context of systemic autoimmune or autoinflammatory diseases.

The Bioethics Committee of the Jagiellonian University Medical College has approved the research (No: 118.6120.41.2023, on 15 June 2023). All procedures adhered to the ethical principles outlined in the Declaration of Helsinki.

### 2.2. Statistical Elaboration

The results were analyzed using IBM SPSS Statistics v. 29.0.0.0 (241). Categorical variables were compared using the exact Fisher test and presented as the number of subjects (*n*) with the percentage of total available data (%). Due to the low number of cases analyzed, we used a non-parametric test to compare continuous data (the Mann–Whitney test), and the results were presented in the text as medians with Q1–Q3 ranges. A significance threshold of two-sided *p*-values below 0.05 was employed for all analyses, and in cases of statistical significance, the *p*-value was bolded in tables.

## 3. Results

### 3.1. Characteristics of Patients with Secondary Hemophagocytic Lymphohistiocytosis

We identified 21 patients who met the criteria described previously and, therefore, were diagnosed with HLH. A summary of the diagnostic process is presented below in Table 3.

Among 21 secondary HLH patients, 12 [57.1%] were women, and the median age in the whole group was 35 (ranging from min 19 to max 67) years. Detailed demographic, clinical, and laboratory characteristics with treatment patterns are provided in Table 4.

We recorded 10 cases (47.6% out of all 21 analyzed) related to rheumatologic diseases, including systemic lupus erythematosus as the most frequent (*n* = 5, 23.8% of all HLH cases). The other five MAS patients represented adult-onset Still’s disease, microscopic polyangiitis, juvenile idiopathic arthritis, idiopathic inflammatory myopathy, and psoriatic arthritis, one patient each (4.8%). Other conditions leading to HLH (*n* = 11, 52.4%) were infections (*n* = 4, 36.4% of all non-MAS cases), with cytomegalovirus (CMV) (*n* = 1, 4.8%), hepatitis B virus (*n* = 1, 4.8%), hepatitis C virus (*n* = 1, 4.8%), Epstein–Barr virus (*n* = 1, 4.8%), and septic shock (*n* = 1, 4.8%), but also lymphoma (*n* = 2, 9.5%) and aceruloplasminemia (*n* = 1, 4.8%). In three cases (14.3% of all HLH), the reason for HLH was unknown. In most cases (*n* = 11, 52.4%), HLH co-occurred with the accompanying illness diagnosis.

Regarding HLH symptoms, the most often were fever (*n* = 21, 100.0%), splenomegaly (*n* = 13, 61.9%), and lymphadenopathy (*n* = 9, 42.9%). In 12 of 18 (66.7%) available data, we confirmed the presence of hemophagocytosis. Interestingly, hemophagocytosis was found in all three cases of HLH triggers. On the other hand, in six patients, hemophagocytosis was not present (cases: No. #2 [HLH trigger: lymphoma], No. #3 [HLH trigger: CMV infection], No. #6 [HLH trigger: septic shock caused by Klebsiella pneumoniae (ESBL-negative, KPC-positive)], No. #13 [HLH trigger: SLE], No. 16 [HLH trigger: lymphoma], and No. 21 [HLH trigger: adult-onset Still’s disease]) (Table 1).

Figure 1 depicts typical phagocytic macrophages (from a single patient) in the bone marrow aspirate (Figure 1A–D).

Most patients had severe clinical manifestations, with a median HScore of 244 and a minimum–maximum range of 208 to 304 points. Considering the treatment pattern, all patients were treated with glucocorticosteroids. Other often used immunosuppressive regimens included cyclosporin (*n* = 12, 57.1%) and intravenous immunoglobulins (*n* = 11, 52.4%). As expected, almost all HLH patients received antibiotics (*n* = 20, 95.2%) and more than half antiviral (*n* = 11, 52.4%) or antifungal (*n* = 12, 57.1%) medications.

In laboratory parameters, HLH patients had leukopenia (neutropenia and lymphopenia) but also anemia and thrombocytopenia. Other stated findings were increased liver enzymes (aspartate and alanine transaminases, lactate dehydrogenase), hypoalbuminemia and hypofibrinogenemia, and high serum levels of triglycerides and ferritin, as well as D-dimer with C-reactive protein. Laboratory findings at HLH diagnosis are summarized in Table 5.

### 3.2. Characteristics of Patients with Hemophagocytic Lymphohistiocytosis Deceased During Hospitalization

Four patients died in acute HLH condition: case No. #2 [HLH trigger: lymphoma], No. #5 [HLH trigger: EBV infection], No. #10 [HLH trigger: unknown], and No. #16 [HLH trigger: lymphoma]). They were characterized by lower white blood cell, neutrophil, and lymphocyte counts but similar demographics and clinical signs. Regarding therapy patterns, they received more aggressive treatment with etoposide as compared to those who survived the acute condition. In detail, patient #2 was diagnosed with splenic lymphoma post-mortem. Patient #5 had a reactivation of the Epstein–Barr virus infection, which occurred in the setting of immunosuppression, likely related to prolonged dexamethasone therapy due to persistent periodic fever. The patient was diagnosed with chronic active EBV infection and HLH, with profound cellular immunodeficiency and multiple opportunistic infections confirming severe immune dysfunction. Patient #16 was diagnosed with B-cell lymphoma—follicular (FL), consistent with grade 2–3A based on the proliferation index (CD20+, CD10+, BCL6+, PAX5+, CD23–, Ki-67 ~50%). Among the remaining 17 patients, we did not record a relapse of HLH symptoms. Still, one patient died after the secondary HLH diagnosis (case No. #20, aceruloplasminemia) due to bleeding to the central nervous system. Interestingly, we documented NK cell dysfunction in all fatal cases; however, the subgroup analysis compared those who survived showed no difference. Below, we have shown a detailed analysis (Table 6).

### 3.3. Characteristics of Patients with Macrophage Activation Syndrome

The 10 MAS patients did not differ in clinics and laboratory findings from the remaining secondary HLH patients, except for slightly lower albumin concentration in peripheral blood in the MAS subgroup (Table 7). Half of MAS cases were related to SLE: three females and two males; the median age was 30 years at the MAS presentation; four out of five patients had lupus nephritis, and all but one had anti-SSA and anti-dsDNA antibodies ever detected. All of them had also hematological manifestations (e.g., leucopenia, lymphopenia, anemia, or thrombocytopenia) ever reported.

## 4. Discussion

In this study, we retrospectively analyzed a group of secondary HLH patients treated in our hospital during the last decade. All patients fulfilled the HLH-2009 criteria and had the HScore > 169 points. The etiology of secondary HLH might be diverse, including infectious triggers, autoimmune disorders, malignancies, drug reactions, or other environmental factors [14,15]. Here, we presented 21 cases of secondary HLH with different etiologies, including 10 patients diagnosed with MAS during the course of rheumatic diseases.

The demographic profile in the analyzed cohort, with a median age of 35 years and a higher prevalence of women, is consistent with prior studies that describe HLH as a syndrome spanning a broad age range and showing no apparent sex predilection in general populations [16,17]. In the literature, the characteristic features of HLH include fever, splenomegaly, and hemophagocytosis. In our data, fever was present in all patients and splenomegaly in 61.9% of them. Hemophagocytosis was detected in 66.7% of those analyzed in this direction. Furthermore, as expected, our secondary HLH patients demonstrated significant laboratory abnormalities, including cytopenias, markedly elevated inflammatory markers (CRP and ferritin), hypertriglyceridemia, and increased liver enzymes. These findings reflect the characteristic hyperinflammatory state and organ dysfunction associated with HLH. Recent investigations highlight the importance of clinical markers such as fever, splenomegaly, and cytopenias as key diagnostic criteria for HLH, confirmed in previous studies [18,19]. Elevated ferritin remains a hallmark of HLH and might serve as a marker of severity [20]. In our study, it was present in almost all patients (*n* = 20/21). On the other hand, one subject with a normal level of ferritin (case No. #9, HLH of unknown reason) was characterized by the presence of typical clinical signs with fever and cytopenia and had bone marrow hemophagocytosis with depleted NK cells. Thus, HLH requires a comprehensive diagnostic approach, and sometimes distinguishing it from sepsis and other hyperinflammatory syndromes remains challenging due to overlapping clinics [21].

In turn, changes in ferritin levels over time can provide valuable insights into the progression of HLH and its response to treatment. Monitoring ferritin dynamics could help identify patients at risk of developing severe forms of the disease and guide therapeutic interventions. Moreover, although in our data, no patient met formal diagnostic criteria for overt DIC, several presented with coagulopathy features, such as thrombocytopenia, elevated D-dimer, and hypofibrinogenemia, that may mimic or overlap with DIC in the context of HLH. Also, NK cell dysfunction is important for diagnosis [22]. In our data, all fatal cases with available data on NK cells had decreased numbers. Other factors, such as the severity of the underlying disease, immune dysregulation, and response to treatment, likely play a significant role in prognosis [23,24]. Unfortunately, we did not have data on clinical status measured with validated scales in patients with connective tissue disorders during the HLH course; thus, we cannot predict the severity of the underlying disease that results in MAS. However, in 4 out of 10 MAS patients (3 cases with SLE and 1 with idiopathic inflammatory myopathy), HLH occurred alongside connective tissue disease manifestations, emphasizing the critical need for early recognition, as this overlap may indicate a severe and potentially fatal disease course.

The HScore, developed to quantify the likelihood of HLH, has emerged as a critical tool in the clinical practice of adult populations [13]. It comprises several features related to HLH and may be related to the disease severity (for details, see Methods section). Nevertheless, we did not find differences in Hscore among specific setups when performing subgroup analyses. Therefore, in our data, Hscore served more as a diagnostic tool than a prognostic one. However, this might be related to the low number of cases. Interestingly, the HScore’s applicability extends beyond traditional HLH settings. Studies validate its effectiveness across diverse populations, including critically ill or oncology patients [25,26]. For instance, in a study by Gualdoni et al. [27], the Hscore was identified as a valuable tool for detecting hyperinflammatory states in SIRS (Systemic Inflammatory Response Syndrome) patients, highlighting its potential to guide urgent therapeutic interventions and predict 30-day survival.

Therapeutic approaches predominantly involve immunosuppressive therapies, including glucocorticosteroids, cyclosporine, and etoposide, as part of the HLH-2004 protocol [28]. However, that protocol has been developed to treat mainly primary HLH, and its goal is to suppress excessive immune system activation and prevent multiorgan failure. In our study, the complete regimen with etoposide was administered only to two patients (cases No. #5 with 230 points in the HScore and No. #10 with 304 points in the HScore, all these two patients had neuropsychiatric symptoms and decreased levels of NK cells) and in both cases the disease was fatal. It highlights the importance of a causal approach in secondary HLH, which is paramount in avoiding a more aggressive immunosuppressive strategy. Therapy of MAS patients in our data was consistent with international guidelines, i.e., glucocorticosteroids being the cornerstone [29]. However, nowadays, EULAR/ACR attempts to develop an evidence-based approach for secondary HLH or during the course of rheumatic diseases (MAS patients) [11], including the use of emerging biologics, such as emapalumab, anakinra, or tocilizumab, as promising drugs in targeting cytokine storm [30]. Baldo et al. [28] summarized current treatment in MAS worldwide, demonstrating that high-dose glucocorticosteroids with IL-1 or IFN-γ inhibitors are good therapeutic options, especially in systemic juvenile idiopathic arthritis-associated MAS. Still, early recognition and differentiation of MAS from sepsis or disease flares are crucial for timely treatment and improved outcomes [31].

Cases of patients with rheumatic diseases in our cohort (47.6%) mirror the risk of MAS in autoimmune conditions [32]. Like other secondary HLH, in clinics, it is characterized by fever, splenomegaly, and hyperferritinemia [33]. In our data, half of MAS patients had SLE. Notably, our hospital center treated approximately 900 SLE patients during the study period, resulting in an estimated MAS prevalence of 0.56%, which is slightly lower than reported in the literature (0.9% to 4.6%) [34]. MAS frequently occurs as systemic inflammation, making it essential to promptly evaluate those with unexplained fever and cytopenias, signs also characteristic of rheumatic diseases [32]. In fact, MAS often presents concurrently with the first signs of the underlying rheumatologic disease, like in most of our patients. In this case, it is associated with a high mortality rate [35]. Moreover, MAS is the major fatal complication of systemic juvenile idiopathic arthritis and shares significant similarities with HLH, emphasizing the need for heightened awareness [8]. Interestingly, in a recent study by Zhu et al. [36], MAS was identified as a common and fatal complication of dermatomyositis patients, strongly associated with rapidly progressive interstitial lung disease and significantly worsening survival outcomes. The same was observed many years ago in the first case of a secondary HLH patient diagnosed in our hospital [37].

Overall, four of our patients died during HLH. All of them had no concomitant rheumatic disease. In laboratory investigations, these patients had lower white blood cells, neutrophils, and lymphocytes than the remaining, which aligns with the current reports. For instance, Bin et al. [38], in an analysis of 116 cases of pediatric predominantly secondary HLH, revealed that severe neutropenia and liver dysfunction were prognostic factors for early death. Moreover, other predictors of fatality included higher interleukin-10, lower albumin, and higher lactate dehydrogenase levels in serum [39], as well as central nervous system involvement and/or presence of thrombocytopenia [40], but also persistent hyperferritinaemia during the time of treatment [41]. Based on the literature, in secondary HLH, mortality is often associated with hematological diseases and infections [42], as in our data. In our study group, two out of four patients who died during the HLH course suffered from lymphoma. Management of malignancy-associated hemophagocytic lymphohistiocytosis remains complex, with corticosteroids and etoposide being the standard treatment, while emerging therapies like ruxolitinib offer the potential for improved outcomes [43]. Ruxolitinib, a JAK1/2 inhibitor, has shown therapeutic potential in HLH by targeting key immune cells involved in hyperinflammation, including macrophages, NK cells, and T cells. Its efficacy is mainly attributed to the inhibition of the JAK/STAT3 signaling pathway, which plays a role in the pathogenesis of HLH [44]. However, treating the underlying disease in secondary HLH is crucial. In fact, Schuster et al. [45] revealed that sepsis, septic shock, liver disease, and hematological malignancies were identified as key differential diagnoses for hyperferritinemia in critically ill adult patients without secondary HLH.

This study has several limitations that we want to acknowledge. It is a retrospective single-center analysis based on a relatively small cohort of patients, resulting in a limited sample size and heterogeneous underlying HLH causes. In three cases with unknown etiology of HLH, the possibility of atypical primary HLH due to undetected or novel genetic mutations cannot be completely excluded despite clinical features highly suggesting secondary HLH. However, every report on secondary HLH is valuable, given the significant challenges in its diagnosis and treatment.

## 5. Conclusions

In conclusion, secondary HLH is a severe inflammatory syndrome that requires prompt diagnosis of itself and a concomitant disease to ensure appropriate management. In our cohort, all MAS patients survived, which suggests that early recognition and optimal rheumatologic treatment may significantly improve outcomes. Given the nonspecific nature of HLH symptoms, a broad differential diagnosis remains essential. However, combining the HLH-2009 criteria and the HScore system strengthens diagnostic confidence and reflects real-world clinical practice. Future studies are needed to refine diagnostic strategies and best treatment options for secondary HLH, including novel biologics.

## Figures and Tables

**Figure 1 medicina-61-00977-f001:**
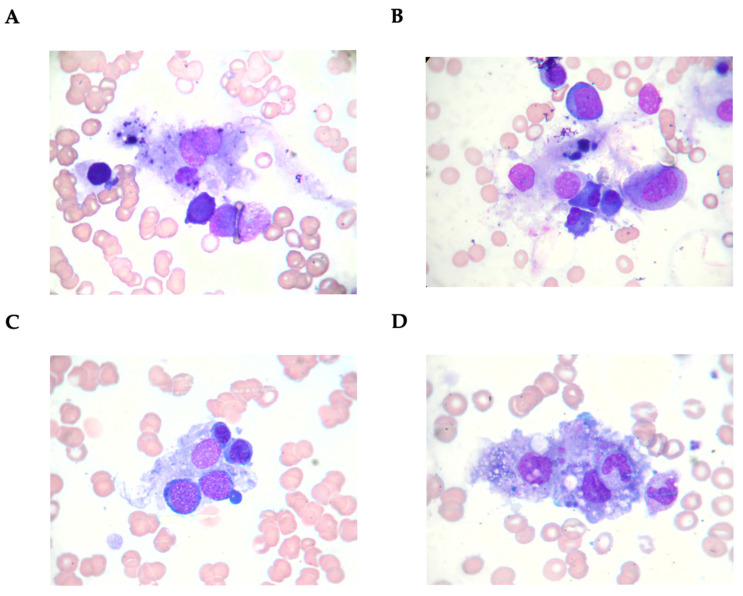
Bone marrow aspirate smear (Wright’s stain, ×1000 magnification). Macrophages laden with erythroblasts/platelets.

**Table 1 medicina-61-00977-t001:** Adapted HLH-2009 diagnostic criteria based on Ref. [12].

Parameter	Values
1. Fever	Body temperature ≥ 38.5 °C
2. Organomegaly (splenomegaly/lymphadenopathy)	Present or absent
3. Bicytopenia or pancytopenia	Neutrophils < 1000/µL
	Hemoglobin < 9 g/dL
	Platelet count < 100 × 10^3^/µL
4. Hypertriglyceridemia	≥ 265 mg/dL
5. Hypofibrinogenemia	≤ 1.8 g/L
6. Hemophagocytosis in bone marrow, liver, spleen, or lymph node	Present or absent
7. Hyperferritinemia	≥ 500 ng/mL
8. Elevated soluble interleukin-2 receptor ^1^	≥ 2400 U/mL
9. Absent or decreased natural killer cell function	< 80 cells/µL

^1^—This test was not performed in our study due to its unavailability in our hospital laboratory and high associated costs.

**Table 2 medicina-61-00977-t002:** Parameters assessed in the Hscore based on Ref. [13].

Parameter	Criteria with Exact Points
Known underlying immunosuppression (human immunodeficiency virus-positive or receiving long-term immunosuppressive therapy [i.e., glucocorticosteroids, cyclosporine, azathioprine])	No → 0 Yes → 18
Temperature, °C	< 38.4 → 0 38.4–39.4 → 33 > 39.4 → 49
Organomegaly	No → 0 Hepatomegaly or splenomegaly → 23 Hepatomegaly and splenomegaly → 38
Number of cytopenias (defined as hemoglobin ≤ 9.2 g/dL and/or white blood cells ≤ 5000/mm^3^ and/or platelets ≤ 110,000/mm^3^)	1 lineage → 0 2 lineages → 24 3 lineages → 34
Ferritin, ng/mL	< 2000 → 0 2000–6000 → 35 > 6000 → 50
Triglycerides, mmol/L	< 1.5 → 0 1.5–4 → 44 > 4 → 64
Fibrinogen, g/L	> 2.5 → 0 ≤ 2.5 → 30
Aspartate transaminase, U/L	< 30 → 0 ≥ 30 → 19
Hemophagocytosis features on bone marrow aspirate	No → 0 Yes → 35

**Table 3 medicina-61-00977-t003:** The characteristics of all patients diagnosed with secondary hemophagocytic lymphohistiocytosis according to the HLH-2009 diagnostic criteria and the HScore.

Criteria	Number of the HLH Case																				
HLH-2009 Criteria	1	2	3	4	5	6	7	8	9	10	11	12	13	14	15	16	17	18	19	20	21
At least 3 of 4:																					
a. fever	Y	Y	Y	Y	Y	Y	Y	Y	Y	Y	Y	Y	Y	Y	Y	Y	Y	Y	Y	Y	Y
b. splenomegaly	N	Y	Y	Y	N	Y	Y	N	Y	Y	Y	N	N	N	Y	N	Y	Y	N	Y	Y
c. cytopenias	Y	Y	Y	Y	Y	Y	Y	Y	Y	Y	Y	Y	Y	Y	Y	Y	Y	Y	Y	Y	Y
d. hepatitis	Y	N	Y	Y	Y	Y	Y	Y	Y	Y	Y	Y	Y	Y	Y	Y	Y	Y	Y	Y	Y
Additionally, at least 1 of 4:																					
a. hemophagocytosis	Y	N	N	Y	Y	N	ND	Y	Y	Y	ND	Y	N	ND	Y	N	Y	Y	Y	Y	N
b. elevated ferritin	Y	Y	Y	Y	Y	Y	Y	Y	N	Y	Y	Y	Y	Y	Y	Y	Y	Y	Y	Y	Y
c. sIL2Ra (age-based)	not done																				
d. NK dysfunction	ND	Y	Y	N	Y	ND	Y	Y	Y	Y	ND	N	ND	Y	ND	ND	ND	N	ND	Y	ND
Other results for diagnosis:																					
a. hypertriglyceridemia	N	Y	Y	Y	Y	Y	Y	Y	Y	Y	Y	Y	Y	Y	Y	Y	ND	N	Y	N	Y
b. hypofibrinogenemia	Y	N	Y	N	Y	Y	N	Y	N	Y	N	Y	Y	N	Y	N	ND	Y	Y	Y	Y
c. hyponatremia	N	N	Y	Y	Y	Y	N	N	Y	Y	N	N	N	Y	N	Y	N	Y	N	N	Y
**HScore**	219	208	218	255	230	269	261	263	223	304	231	282	271	214	271	244	211	258	281	209	217
**Other clinical data**																					
**Age at the time of HLH, years**	35	64	62	25	56	20	25	53	23	19	22	38	36	44	30	67	32	19	39	29	54
**Sex (female/male)**	M	M	F	M	F	M	F	F	F	M	F	F	M	F	F	F	F	M	F	M	M
**Etiology** **of the secondary HLH**	SLE	lymphoma	cytomegalovirus infection	hepatitis B virus	Epstein Barr virus	septic shock	SLE	IIM	unknown	unknown	SLE	hepatitis C virus	SLE	microscopic polyangiitis	SLE	lymphoma	juvenile idiopathic arthritis	unknown	psoriatic arthritis	aceruloplasminemia	adult-onset Still’s disease
**Outcome during the HLH**	S	D	S	S	D	S	S	S	S	D	S	S	S	S	S	D	S	S	S	S	S

Abbreviations: Y—yes, N—no, HLH—hemophagocytic lymphohistiocytosis, NK—natural killer, ND—no data, sIL2Ra—soluble interleukin-2 receptor alpha, S—survival case, D—death (fatal case), F—female sex, M—male sex, SLE—systemic lupus erythematosus, IIM—idiopathic inflammatory myopathy.

**Table 4 medicina-61-00977-t004:** Demographic, clinical, and treatment characteristics of all enrolled patients.

Parameter	Secondary HLH Patients *n* = 21
**Demographic factors**	
Age at the secondary HLH diagnosis, years	35.0 (24.0–53.5)
Sex, female, *n* (%)	12 (57.1%)
**Clinical characteristics**	
Age at the accompanying illness diagnosis, years	32.5 (24.8–53.8)
Time from illness to secondary HLH diagnosis, years	0.0 (0.0–4.3)
Splenomegaly, *n* (%)	13 (61.9%)
Lymphadenopathy, *n* (%)	9 (42.9%)
Neuropsychiatric symptoms, *n* (%)	8 (38.1%)
Fever, *n* (%)	21 (100.0%)
Disseminated intravascular coagulation, *n* (%)	1 (4.8%)
Hemophagocytosis ^1^, *n* (%)	12 (66.7%)
Hypofibrinogenemia ^2^, *n* (%)	13 (65.0%)
Death due to HLH, *n* (%)	4 (19.0%)
HScore, points	244 (218–270)
**Treatment pattern**	
Glucocorticosteroids, *n* (%)	21 (100.0%)
Immunoglobulin, *n* (%)	11 (52.4%)
Rituximab, *n* (%)	2 (9.5%)
Cyclosporin, *n* (%)	12 (57.1%)
Etoposide, *n* (%)	2 (9.5%)
Antibiotics, *n* (%)	20 (95.2%)
Antiviral drugs, *n* (%)	11 (52.4%)
Antifungal drugs, *n* (%)	12 (57.1%)

Categorical variables are presented as numbers with percentages, whereas continuous variables are presented as medians and Q1–Q3 ranges. ^1^—the results were available for 18 out of 21 patients (85.7%). ^2^—the results were available for 20 out of 21 patients (95.2%). Abbreviations: *n*—number; HLH—hemophagocytic lymphohistiocytosis.

**Table 5 medicina-61-00977-t005:** A summary of laboratory findings in all enrolled secondary HLH patients.

Parameter	Secondary HLH Patients *n* = 21	Reference Ranges
White blood cells, 10^3^/µL	2.16 (1.15–5.48)	4.0–10.0
Neutrophils, 10^3^/µL	1.11 (0.40–2.92)	1.8–7.7
Lymphocytes, 10^3^/µL	0.55 (0.39–1.09)	1.0–4.5
Lymphocytes NK, 10^3^/µL	0.047 (0.018–0.089)	0.080–0.350
Hemoglobin, g/dL	7.8 (6.1–9.6)	14.0–18.0
Blood platelets, 10^3^/µL	81 (32–116)	140–440
Aspartate transaminase, U/L	127 (92–289)	10–50
Alanine transaminase, U/L	109 (88–263)	10–50
Lactate dehydrogenase, IU/L	1372 (591–2009)	135–225
Albumin, g/L	26.3 (20.0–31.6)	35.0–52.0
Triglycerides, mmol/L	4.53 (3.11–6.76)	<2.26
Fibrinogen, g/L	1.31 (0.78–2.12)	1.8–3.5
Ferritin, ng/mL	14,305 (5632–30,342)	13–400
D-dimer, mg/L	3.88 (2.78–35.00)	<0.55
C-reactive protein, mg/L	224.3 (44.5–277.5)	<5.0

Continuous variables are presented as medians and Q1–Q3 ranges. Abbreviations: *n*—number; HLH—hemophagocytic lymphohistiocytosis.

**Table 6 medicina-61-00977-t006:** Differences in characteristics regarding the main outcome, fatal cases vs. survived patients.

Parameter	Fatal HLH Cases *n* = 4	Survived HLH Patients *n* = 17	*p*-Value
**Demographic factors**			
Age at HLH diagnosis, years	60.0 (28.3–66.3)	32.0 (24.0–41.5)	0.14
Sex, female, *n* (%)	2 (50.0%)	10 (58.8%)	1.00
**Clinical characteristics**			
Macrophage activation syndrome, *n* (%)	0 (0.0%)	10 (58.8%)	0.09
Splenomegaly, *n* (%)	2 (50.0%)	11 (64.7%)	0.62
Lymphadenopathy, *n* (%)	2 (50.0%)	7 (41.2%)	1.00
Neuropsychiatric symptoms, *n* (%)	3 (75.0%)	5 (29.4%)	0.25
Fever, *n* (%)	4 (100.0%)	17 (100.0%)	NA
Disseminated intravascular coagulation, *n* (%)	0 (0.0%)	1 (5.9%)	1.00
Hemophagocytosis, *n* (%)	2 (50.0%)	10 (58.8%)	1.00
Hscore, points	237 (214–289)	255 (218–270)	0.90
**Treatment pattern**			
Glucocorticosteroids, *n* (%)	4 (100.0%)	17 (100.0%)	NA
Immunoglobulin, *n* (%)	2 (50.0%)	9 (52.9%)	1.00
Rituximab, *n* (%)	0 (0.0%)	2 (11.8%)	1.00
Cyclosporin, *n* (%)	3 (75.0%)	9 (52.9%)	0.60
Etoposide, *n* (%)	2 (50.0%)	0 (0.0%)	**0.029**
Antibiotics, *n* (%)	4 (100.0%)	16 (94.1%)	1.00
Antiviral drugs, *n* (%)	3 (75.0%)	8 (47.1%)	0.59
Antifungal drugs, *n* (%)	4 (100.0%)	8 (47.1%)	0.10
**Laboratory findings**			
White blood cells, 10^3^/µL	0.77 (0.31–1.64)	2.69 (1.60–5.87)	**0.031**
Neutrophils, 10^3^/µL	0.23 (0.02–0.78)	2.09 (0.90–4.70)	**0.020**
Lymphocytes, 10^3^/µL	0.27 (0.20–0.50)	0.66 (0.42–1.23)	**0.039**
Lymphocytes NK, 10^3^/µL	0.010 (0.002–0.055)	0.060 (0.033–0.206)	0.10
Hemoglobin, g/dL	6.7 (5.5–8.5)	8.1 (6.6–10.1)	0.36
Blood platelets, 10^3^/µL	14 (7–99)	81 (52–116)	0.12
Aspartate transaminase, U/L	147 (52–308)	121 (92–290)	0.97
Alanine transaminase, U/L	248 (91–562)	105 (88–240)	0.44
Lactate dehydrogenase, IU/L	1781 (1414–5352)	1151 (498–1848)	0.10
Albumin, g/L	25.5 (24.3–26.2)	29.1 (20.0–33.1)	0.53
Triglycerides, mmol/L	3.99 (3.20–4.68)	4.65 (2.56–7.16)	0.49
Fibrinogen, g/L	2.00 (1.00–3.83)	1.31 (0.69–2.07)	0.44
Ferritin, ng/mL	9385 (4116–22,657)	20,133 (7411–34,836)	0.36
D-dimer, mg/L	2.60 (2.44–2.60)	6.52 (2.88–35.10)	0.15
C-reactive protein, mg/L	236.1 (111.8–314.4)	199.0 (21.0–277.5)	0.49

Categorical variables are presented as numbers with percentages, whereas continuous variables are presented as medians and Q1–Q3 ranges. Statistically significant differences are bolded (Fisher’s exact test or Mann–Whitney test). Abbreviations: *n*—number, HLH—hemophagocytic lymphohistiocytosis, NA—not applicable.

**Table 7 medicina-61-00977-t007:** Differences in characteristics between macrophage activation syndrome and remaining secondary hemophagocytic histiocytosis patients.

Parameter	MAS Patients *n* = 10	Non-MAS HLH Patients *n* = 11	*p*-Value
**Demographic factors**			
Age during HLH, years	35.5 (28.8–46.3)	29.0 (20.0–62.0)	0.76
Sex, female, *n* (%)	7 (70.0%)	5 (45.5%)	0.39
**Clinical characteristics**			
Death during hospitalization, *n* (%)	0 (0.0%)	4 (36.4%)	0.09
Splenomegaly, *n* (%)	5 (50.0%)	8 (72.7%)	0.39
Lymphadenopathy, *n* (%)	5 (50.0%)	4 (36.4%)	0.67
Neuropsychiatric symptoms, *n* (%)	3 (30.0%)	5 (45.5%)	0.66
Fever, *n* (%)	10 (100.0%)	11 (100.0%)	NA
Disseminated intravascular coagulation, *n* (%)	0 (0.0%)	1 (9.1%)	1.00
Hemophagocytosis, *n* (%)	5 (50.0%)	7 (63.6%)	0.67
HScore, points	246 (216–271)	244 (218–269)	0.92
**Treatment pattern**			
Glucocorticosteroids, *n* (%)	10 (100.0%)	11 (100.0%)	NA
Immunoglobulin, *n* (%)	4 (40.0%)	7 (63.6%)	0.40
Rituximab, *n* (%)	2 (20.0%)	0 (0.0%)	0.21
Cyclosporin, *n* (%)	7 (70.0%)	5 (45.5%)	0.39
Etoposide, *n* (%)	0 (0.0%)	2 (18.2%)	0.48
Antibiotics, *n* (%)	10 (100.0%)	10 (90.9%)	1.00
Antiviral drugs, *n* (%)	4 (40.0%)	7 (63.6%)	0.40
Antifungal drugs, *n* (%)	5 (50.0%)	7 (63.6%)	0.67
**Laboratory findings**			
White blood cells, 10^3^/µL	3.47 (0.95–6.44)	1.79 (1.19–2.69)	0.31
Neutrophils, 10^3^/µL	2.34 (0.48–4.83)	1.00 (0.32–1.44)	0.21
Lymphocytes, 10^3^/µL	0.53 (0.40–1.53)	0.56 (0.33–0.90)	1.00
Lymphocytes NK, *n*/μL	0.036 (0.014–0.036)	0.055 (0.020–0.206)	0.73
Hemoglobin, g/dL	7.7 (4.9–10.6)	7.8 (6.3–9.5)	0.71
Blood platelets, 10^3^/µL	81 (64–119)	57 (13–116)	0.39
Aspartate transaminase, U/L	149 (112–253)	104 (50–362)	0.31
Alanine transaminase, U/L	92 (76–222)	118 (97–375)	0.30
Lactate dehydrogenase, IU/L	1483 (1061–2675)	1027 (469–1700)	0.28
Albumin, g/L	22.0 (18.2–28.8)	29.1 (25.0–34.9)	**0.041**
Triglycerides, mmol/L	4.69 (2.82–7.70)	4.30 (3.04–5.10)	0.37
Fibrinogen, g/L	1.32 (0.59–2.11)	1.30 (1.00–3.00)	0.60
Ferritin, ng/mL	21,682 (11,591–33,937)	11,708 (4000–25,441)	0.17
D-dimer, mg/L	10.03 (3.11–35.20)	2.83 (1.50–28.30)	0.08
C-reactive protein, mg/L	181.0 (12.2–259.2)	235.0 (71.8–288.0)	0.30

Categorical variables are presented as numbers with percentages, whereas continuous variables are presented as medians and Q1–Q3 ranges. Statistically significant differences are bolded (Mann–Whitney test). Abbreviations: *n*—number; HLH—hemophagocytic lymphohistiocytosis, NA—not applicable.

## Data Availability

The data presented in this study are available upon reasonable request from the corresponding author.

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
