# Peer review of "Characteristics of 21 Patients with Secondary Hemophagocytic Lymphohistiocytosis—Insights from a Single-Center Retrospective Study"

_medicina, 2025, doi:10.3390/medicina61060977_

Round 1

Reviewer 1 Report

Comments and Suggestions for Authors

I reviewed a manuscript entitled “Characteristics of 21 Patients with Secondary Hemophagocytic Lymphohistiocytosis – Insights from a Single-Center Retrospective Study” written by Dr. Radosław Dziedzic et al., which reported a clinical characteristic of secondary hemophagocytic lymphohistiocytosis (HLH). And more they analyzed a concomitancy of macrophage activation syndrome (MAS). This retrospective clinical research presented a possibility of better prognosis in HLH accompanied with MAS compared to it without MAS. This report was interesting both for hematologists and rheumatologists. I have some clinical questions.

Major comments

  1. The authors mentioned macrophage activation syndrome (MAS) is a by-standing clinicopathology in hemophagocytic lymphohistiocytosis (HLH) in the Introduction part (line 53). Much more you used mixed the terminology “HLH/MAS” also in the Introduction (line 72). Please define the common findings and different points of HLH and MAS. Please clarify what you would like to compare between these two syndromes.
  2. You compared the fatal HLH cases and non-fatal HH cases, shown in Table 4. Did you find any contributing risk factors for death in your study cohort? I hope you can perform multivariate analysis to identify the risk.
  3. Table 5 was excellent to draw your research result and speculation, I believe. What was relevant difference between MAS/HLH and non-MAS/HLH cases. And please teach us and leaders what is an essential property of better prognosis in MAS patient rather than non-MAS patients.

Minor comments

  1. Please define the diagnostic criteria of MAS you adopted in your study.
  2. The test for NK dysfunction is not routine clinical practice. Please clarify the method of the test.
  3. The Figure 1 indicated four pictures from each four case? Or four typical pictures from a single case?
  4. Were there any patient with “hypo”fibrinogenemia?
  5. How was a complication of disseminated intravascular coagulation (DIC) in your study cohort. If any, please specify the cases and details.

Author Response

Reviewer 1

I reviewed a manuscript entitled “Characteristics of 21 Patients with Secondary Hemophagocytic Lymphohistiocytosis – Insights from a Single-Center Retrospective Study” written by Dr. Radosław Dziedzic et al., which reported a clinical characteristic of secondary hemophagocytic lymphohistiocytosis (HLH). And more they analyzed a concomitancy of macrophage activation syndrome (MAS). This retrospective clinical research presented a possibility of better prognosis in HLH accompanied with MAS compared to it without MAS. This report was interesting both for hematologists and rheumatologists. I have some clinical questions.

General response to comments:

The authors sincerely appreciate Reviewer 1 for the thorough evaluation of our work and for offering insightful suggestions that have enhanced the quality of our manuscript. Below, we have provided point-by-point responses to the comments.

Major comments

  1. The authors mentioned macrophage activation syndrome (MAS) is a by-standing clinicopathology in hemophagocytic lymphohistiocytosis (HLH) in the Introduction part (line 53). Much more you used mixed the terminology “HLH/MAS” also in the Introduction (line 72). Please define the common findings and different points of HLH and MAS. Please clarify what you would like to compare between these two syndromes.

Response:

Thank you for this valuable comment. Our primary objective was to analyze the clinical course of secondary hemophagocytic lymphohistiocytosis (HLH) in patients treated in the last decade in our hospital. We also acknowledge that macrophage activation syndrome (MAS) is a specific form of secondary HLH that occurs in the context of rheumatic diseases, most commonly systemic juvenile idiopathic arthritis or Still’s disease. To avoid confusion, we have revised the terminology throughout the manuscript, replacing “HLH/MAS” with either “ secondary HLH” or “MAS” as appropriate. We aimed to investigate whether clinical and laboratory parameters differ between MAS and non-MAS secondary HLH patients to understand the pathophysiological distinctions and potential implications for diagnosis and treatment.

  1. You compared the fatal HLH cases and non-fatal HH cases, shown in Table 4. Did you find any contributing risk factors for death in your study cohort? I hope you can perform multivariate analysis to identify the risk.

Response:

Thank you very much for this insightful comment. We analyzed demographics, clinics, and laboratory parameters to identify potential risk factors for fatality. As shown in Table 6, patients in the fatal secondary HLH subgroup had significantly lower white blood cell, neutrophil, and lymphocyte counts (p < 0.05 for all), which may suggest that deeper cytopenia could be associated with poorer prognosis, as also observed in other papers. Additionally, those patients were also more frequently treated with etoposide, which aligns with current recommendations for managing severe or refractory HLH. We have discussed that issue in the Discussion section.

Unfortunately, a more sophisticated statistical approach was not possible in such a limited number of cases. Nevertheless, we believe that even this data from a small cohort provides valuable clinical insights. For instance, no fatality was observed in properly treated rheumatic MAS patients.

  1. Table 5 was excellent to draw your research result and speculation, I believe. What was relevant difference between MAS/HLH and non-MAS/HLH cases. And please teach us and leaders what is an essential property of better prognosis in MAS patient rather than non-MAS patients.

Response:

Thank you very much for your kind words regarding that Table. In our analysis, we did not observe important differences between MAS and non-MAS secondary HLH subgroups in terms of demographics or clinics, besides that fatality was documented only in secondary HLH patients other than MAS. In rheumatic diseases, initiating appropriate immunosuppressive therapy is essential in preventing the progression to MAS. The relatively better prognosis observed in MAS patients may be partially attributed to heightened awareness among rheumatologists, routine monitoring for MAS-specific warning signs (such as clinical symptoms and laboratory parameters like rising ferritin or cytopenias), and timely escalation of therapy. That underlines the importance of multidisciplinary collaboration and disease-specific management protocols in improving outcomes in this distinct secondary HLH subgroup. We discussed that issue in the Discussion section of our improved paper.

Minor comments

  1. Please define the diagnostic criteria of MAS you adopted in your study.

Response:

Thank you for the comment. In our study, secondary HLH was diagnosed based on the HLH-2009 criteria and the HScore above the diagnostic threshold (> 169 points), which are widely accepted tools for identifying hemophagocytic syndromes. In patients with underlying rheumatic diseases, cases fulfilling HLH diagnostic criteria were further classified as MAS (macrophage activation syndrome) following the current understanding that MAS represents a subtype of secondary HLH occurring in the context of systemic autoimmune or autoinflammatory disorders. We have added this definition in the Methods section.

  1. The test for NK dysfunction is not routine clinical practice. Please clarify the method of the test.

Response:

Thank you very much for your comment. NK cells were identified in peripheral blood by flow cytometry as CD3CD56 lymphocytes. Results are presented as percentages and absolute counts. We have provided this information in the Methods section.

  1. The Figure 1 indicated four pictures from each four case? Or four typical pictures from a single case?

Response:

The images presented in Figure 1 depict representative histopathological findings consistent with HLH, taken from a single patient included in our study cohort. These photos are intended to illustrate typical features observed in bone marrow during active secondary HLH.

  1. Were there any patient with “hypo”fibrinogenemia?

Response:

Hypofibrinogenemia (defined as a fibrinogen level below 1.8 g/L as stated in the Methods section) was observed in 13 out of 20 patients (with fibrinogen data unavailable for one patient, case No. #17). Detailed clinical and laboratory characteristics have been provided in Table 3, including patients with hypofibrinogenemia. Additionally, we have updated Table 4 to include the number of patients with hypofibrinogenemia for clarity.

  1. How was a complication of disseminated intravascular coagulation (DIC) in your study cohort. If any, please specify the cases and details.

Response:

In our study cohort, we did not diagnose any patient with overt disseminated intravascular coagulation (DIC) based on established scoring systems such as the ISTH DIC score. However, several patients presented with laboratory features suggestive of coagulopathy, including thrombocytopenia, elevated D-dimer, and hypofibrinogenemia, which are commonly seen in HLH and may mimic or overlap with DIC. We have now added a comment regarding this observation in the Discussion section to highlight the diagnostic challenge in those patients.

We hope that the current version of the manuscript is suitable for publication. Once again, thank you very much for your comments and suggestions.

Reviewer 2 Report

Comments and Suggestions for Authors

This manuscript presents a retrospective single-center study of 21 patients diagnosed with secondary hemophagocytic lymphohistiocytosis (HLH), including a subset with macrophage activation syndrome (MAS). The authors provide a comprehensive overview of the clinical presentation, laboratory parameters, treatment patterns, and outcomes over an 11-year period.

The topic is of high clinical relevance, especially given the rarity and severity of HLH, and the lack of real-world data on adult cases. The manuscript is generally well written and structured, and the results are presented clearly. The inclusion of subgroup analyses (fatal vs. survived, MAS vs. non-MAS) is particularly valuable.

However, some clarifications and improvements are recommended to enhance the manuscript’s scientific rigor and readability.

  1. Please clarify how many of the total patients number were screened and how the final 21 patients were selected for this pariculr study. Were any patients excluded due to missing data or any othere reason or not meeting criterions?
  2. NK cell values were available in most patients, but did you measure that? Please describe the method? Was that most probably flow cytometry or something else?
  3. Highlight the statist. sig. p-values.
  4. As ferritin is emphasized as a diagnostic and prognostic marker, do you have available data on its change over time durng the hospitalization? That would be very benefitial. If not, please discuss its importance in the discussion section.
  5. Try not to use phrases like „all but one“. It is better to give exact numbers.

With the suggested clarifications and improvements, the manuscript will make a valuable contribution to the literature on secondary HLH.

Author Response

Reviewer 2

This manuscript presents a retrospective single-center study of 21 patients diagnosed with secondary hemophagocytic lymphohistiocytosis (HLH), including a subset with macrophage activation syndrome (MAS). The authors provide a comprehensive overview of the clinical presentation, laboratory parameters, treatment patterns, and outcomes over an 11-year period.

The topic is of high clinical relevance, especially given the rarity and severity of HLH, and the lack of real-world data on adult cases. The manuscript is generally well written and structured, and the results are presented clearly. The inclusion of subgroup analyses (fatal vs. survived, MAS vs. non-MAS) is particularly valuable.

However, some clarifications and improvements are recommended to enhance the manuscript’s scientific rigor and readability.

General response to comments:

The authors would like to thank Reviewer 2 for the thorough evaluation of our work and for offering insightful comments that helped us substantially improve the manuscript. Below, we have provided point-by-point responses to the comments.

  1. Please clarify how many of the total patients number were screened and how the final 21 patients were selected for this pariculr study. Were any patients excluded due to missing data or any othere reason or not meeting criterions?

Response:

Thank you very much for your comment. For this study, we included all patients with secondary HLH treated in our hospital during the last decade. They must fulfill the diagnostic criteria for HLH according to the HLH-2009 criteria and have an HScore greater than 169 points, which is considered the diagnostic threshold. Initially, 46 patients were identified based on ICD-10 coding. However, several had alternative final diagnoses or did not meet the required diagnostic criteria. Therefore, we focused on a clearly defined and diagnostically confirmed cohort of 21 patients to ensure the reliability and interpretability of our findings.

  1. NK cell values were available in most patients, but did you measure that? Please describe the method? Was that most probably flow cytometry or something else?

Response:

Thank you very much for your comment. NK cells were identified in peripheral blood by flow cytometry as CD3CD56 lymphocytes. Results were presented as percentages and absolute counts. We have provided this information in the Methods section.

  1. Highlight the statist. sig. p-values.

Response:

Thank you very much for your suggestion. The manuscript has been improved accordingly.

  1. As ferritin is emphasized as a diagnostic and prognostic marker, do you have available data on its change over time durng the hospitalization? That would be very benefitial. If not, please discuss its importance in the discussion section.

Response:

Thank you very much for your suggestion. Our analysis did not include changes in laboratory parameters over time. We have focused on clinical assessment at a specific time at secondary HLH diagnosis. However, changes in ferritin levels over time can serve as an important diagnostic and prognostic marker in this disease. We have added a paragraph in the Discussion section to develop this issue.

  1. Try not to use phrases like „all but one“. It is better to give exact numbers.

Response:

Thank you for your remark. We have improved the manuscript, as suggested.

With the suggested clarifications and improvements, the manuscript will make a valuable contribution to the literature on secondary HLH.

We hope that the current version of the manuscript is suitable for publication. Once again, thank you very much for your valuable comments and suggestions.

Reviewer 3 Report

Comments and Suggestions for Authors

Secondary HLH is an uncommon disease; therefore, the description of individual clinical cases of this disease may be of interest to readers of the journal Medicine. A general positive evaluation of the results of this study is warranted, although a few comments are warranted.

(1) Abstract, Introduction. The main trigger factors of HLH are not only rheumatic diseases, but also infectious diseases, immunodeficiency, and certain medications.

(2) Patients and Methods. It is recommended that the HScore criteria be provided in its entirety within the form of a table, or alternatively, that a link be provided to an open access publication, such as [PMID: 33867732; PMC8012436]. The link provided by the authors [13] is not an open access article. Furthermore, the methods described do not align with the title of the subsection (2.1. Study population).

(3) Discussion, lines 173–175. Mast cells are not the main link in the action of Ruxolitinib in HLH therapy, given the multifaceted nature of this JAK1,2 inhibitor, which encompasses actions on macrophages, NK cells, and T cells [PMID: 33664745; PMCID: PMC7923355]. The JAK2/STAT5 signaling pathway plays a pivotal role in regulating mast cell growth and survival, as previously documented in reference [44]. However, for the treatment of HLH, suppression of the JAK/STAT3 pathway in the aforementioned cells is of greater importance.

(4) The authors compared the HScore and HLH-2009 criteria; however, the conclusions section does not summarize these comparisons, including the strengths and weaknesses of these systems, the most preferred points of their application, etc.

(5) References: According to MDPI rules, a period is placed after abbreviated words in the titles of journals, for example, Am. J. Blood Res.

Author Response

Reviewer 3

Secondary HLH is an uncommon disease; therefore, the description of individual clinical cases of this disease may be of interest to readers of the journal Medicine. A general positive evaluation of the results of this study is warranted, although a few comments are warranted.

General response to comments:

The authors sincerely appreciate Reviewer 3 for the thorough evaluation of our work and for offering insightful suggestions that have enhanced the quality of our manuscript. Below, we have provided point-by-point responses to the comments.

(1) Abstract, Introduction. The main trigger factors of HLH are not only rheumatic diseases, but also infectious diseases, immunodeficiency, and certain medications.

Response:

Thank you very much for your comment. We improved our manuscript accordingly.

(2) Patients and Methods. It is recommended that the HScore criteria be provided in its entirety within the form of a table, or alternatively, that a link be provided to an open access publication, such as [PMID: 33867732; PMC80124360/0/00 0:00:00 AM]. The link provided by the authors [13] is not an open access article. Furthermore, the methods described do not align with the title of the subsection (2.1. Study population).

Response:

Thank you very much for your insightful comment. We agree with the Reviewer’s remark and have provided two new tables: one presenting the HLH-2009 diagnostic criteria in detail (Table 1), and another outlining the parameters and scoring system used in the HScore (Table 2).

(3) Discussion, lines 173–175. Mast cells are not the main link in the action of Ruxolitinib in HLH therapy, given the multifaceted nature of this JAK1,2 inhibitor, which encompasses actions on macrophages, NK cells, and T cells [PMID: 33664745; PMCID: PMC7923355]. The JAK2/STAT5 signaling pathway plays a pivotal role in regulating mast cell growth and survival, as previously documented in reference [44]. However, for the treatment of HLH, suppression of the JAK/STAT3 pathway in the aforementioned cells is of greater importance.

Response:

Thank you very much for your valuable comment. We have revised the text accordingly to reflect better the current understanding of ruxolitinib’s mechanism of action in HLH treatment.

(4) The authors compared the HScore and HLH-2009 criteria; however, the conclusions section does not summarize these comparisons, including the strengths and weaknesses of these systems, the most preferred points of their application, etc.

Response:

Thank you for this important observation. From a clinical standpoint, using both diagnostic tools enhances the credibility of an HLH diagnosis, especially given the complexity and heterogeneity of its presentation. Each scoring system has its own strengths and limitations; therefore, their combined application provides a more comprehensive assessment. Additionally, due to the overlapping features of HLH with other hyperinflammatory conditions, it is crucial always to conduct a broad differential diagnosis. We have revised the Conclusions section to include these points and clarify the practical implications of using both criteria in clinical practice.

(5) References: According to MDPI rules, a period is placed after abbreviated words in the titles of journals, for example, Am. J. Blood Res.

Response:

Thank you very much for your guidance. We have improved the citation style as suggested.

We hope that the current version of the manuscript is suitable for publication. Once again, thank you very much for your valuable comments and suggestions.